# The Moderating Role of the School Context on the Effects of the Healthy Primary School of the Future

**DOI:** 10.3390/ijerph16132432

**Published:** 2019-07-09

**Authors:** Nina Bartelink, Patricia van Assema, Maria Jansen, Hans Savelberg, Stef Kremers

**Affiliations:** 1Department of Health Promotion, Care and Public Health Research Institute (CAPHRI), Maastricht University, P.O. Box 616, 6200 MD Maastricht, The Netherlands; 2Department of Health Promotion, School of Nutrition and Translational Research in Metabolism (NUTRIM), Maastricht University, P.O. Box 616, 6200 MD Maastricht, The Netherlands; 3Academic Collaborative Centre for Public Health Limburg, Public Health Services, P.O. Box 33, 6400 AA Heerlen, The Netherlands; 4Department of Health Services Research, Care and Public Health Research Institute (CAPHRI), Maastricht University, P.O. Box 616, 6200 MD Maastricht, The Netherlands; 5Department of Nutritional and Movement Sciences, Nutrition and Translational Research Institute Maastricht (NUTRIM), Maastricht University, P.O. Box 616, 6200 MD Maastricht, The Netherlands

**Keywords:** complex adaptive systems, health promoting schools, mixed-methods design, moderators, qualitative comparison, quasi-experimental design, school context

## Abstract

*Background*: The current study investigated the moderating role of the school context on the effects of a Dutch health promoting school initiative on children’s health and health behaviors. *Methods*: The study used a mixed-methods design. The school context (*n* = 4) was assessed by the characteristics of the school population, teacher’s health-promoting (HP) practices, implementers’ perceived barriers, school’s HP elements, and dominating organizational issues. Outcomes included objectively assessed BMI z-scores and physical activity (PA), and parent and child-reported dietary intake. Analyses included linear mixed models (four intervention schools versus four control schools), and qualitative comparisons between intervention schools with similar HP changes. *Results*: Effects on outcomes varied considerably across schools (e.g., range in effect size on light PA of 0.01–0.26). Potentially moderating contextual aspects were the child’s socioeconomic background and baseline health behaviors; practices and perceived barriers of employees; and organizational issues at a school level. *Conclusions*: Similar HP changes lead to different outcomes across schools due to differences in the school context. The adoption of a complex adaptive systems perspective contributes to a better understanding of the variation in effects and it can provide insight on which contextual aspects to focus on or intervene in to optimize the effects of HP initiatives.

## 1. Introduction

Promoting healthy behaviors at an early age helps to improve children’s health and their academic achievements [1,2]. This may lead to improved health later in life and reduce the socioeconomic inequity in both health and academic achievement [1,2]. Schools have the potential to support children in improving their health behaviors [3,4,5]. However, school health promotion is often characterized globally by fragmentation, relatively low priority, and a lack of coordination [6,7]. The Health Promoting School framework, as defined by the World Health Organization, aims for a whole-school approach and it focuses on embedding health and well-being in the curriculum, creating healthy social and physical environments, and engaging with parents and the wider community [8]. However, even though this strategy to integrate health promotion into the whole school system is promising, suboptimal results are often observed, due to, among other things, challenges regarding the implementation of specific health-promoting (HP) changes as part of this school-wide change and how to create a meaningful impact [9,10,11,12,13,14]. To understand these challenges, the suggestion has been made to consider schools as complex adaptive systems [15,16]. A complex adaptive system can be described as a system that consists of many interacting components and has the capability to self-organize and adapt. The system’s behavior is typically non-linear, not easily controlled or predicted, and it tends to self-organize to a state of stability [12,14,17,18,19,20]. This means that each complex adaptive system acts in a unique way and can react differently to changes, since each one has its own context. Embracing this perspective of considering schools as complex adaptive systems means that it depends on the specific school context whether a specific HP change fits in a school, and that in each school, the implementation process of a specific HP change is different [21,22]. It also means that even when similar HP changes are achieved, these can have different effects across schools as the changes may be moderated by the unique context of the school [22,23,24]. Several studies have examined the role of the school context, but mainly focused on its interaction with HP changes during the implementation process [12,25]. The focus of this study was to examine the moderating role of the school context on the effects of HP changes when implementation was comparable between schools. This should contribute to a better understanding of the variation in effects of HP initiatives that is often found between schools [26].

School context is defined as the specific circumstances and characteristics of each school, which relates to the social, political, economic, and physical environment; the characteristics, behaviors, wishes, and needs of the people in the school; the wider community in which the school is located; as well as the history and organization of the school [27,28]. This definition shows that a school context consists of many different aspects. Previous studies have shown that some specific aspects might be of importance for school health promotion efforts. They include: characteristics of the school population (demographics, current health behaviors, health and well-being) [26]; HP practices of the teachers [12]; perceived barriers for implementation of HP initiatives, which can be categorized into barriers related to the users, the innovation, the support, the organization, and the socio-political environment [29]; current HP elements in the school (school routine, policy, education, and environment) [8]; and dominating organizational issues, e.g., merger process [26].

In a previous study, we developed a program theory (Figure 1) to visualize the interaction between the school context and the ‘Healthy Primary School of the Future’ (HPSF), which is a Dutch health promoting school initiative [28]. Part of this program theory concerns the moderating role of the school context on the effects of HPSF, as visualized by the moderator arrow in the top right of the model. The aim of the study was to explore the moderating role of the school context on the effects of HPSF among four primary schools (aged 4 to 12). Three research questions were formulated: (1) How did the school contexts differ from each other? (2) What are the effects of HPSF in each school on children’s BMI z-score and their dietary and PA behaviors? and (3) Which aspects of the context relate to larger favorable effects of HPSF?

## 2. Materials and Methods

### 2.1. The Healthy Primary School of the Future

HPSF is a Dutch initiative that aims to sustainably integrate health and well-being within the whole school system. Three cooperating organizations developed the idea for HPSF: the regional educational board ‘Movare’, the regional Public Health Services, and Maastricht University [30]. HPSF is based on the principles of the Health Promoting School framework and intends to establish a broad collaboration between the school, parents, and external partners, to develop and implement HP changes in the whole school system, e.g., the school’s physical and social environment, its health policy, education, and routines [30,31]. On top of the Health Promoting School framework, the initiative aims to create some form of positive disruption in the school, by initiating two changes top-down: (1) a free healthy lunch each day and (2) structured PA and cultural sessions after lunch, both led by external pedagogical employees provided by childcare organizations. These two changes should create momentum for bottom-up processes to implement additional HP changes [31]. Each school selected a teacher as the school coordinator, who managed HPSF in their school. Overarching the schools, the HPSF initiative was led by a project leader from Movare and an executive board with representatives from the three collaborating organizations, including the project leader. A project team was created with representatives of all the partners involved: the four schools, Movare, regional Public Health Services, Maastricht University, childcare organizations, sports and leisure organizations, a caterer, and the Limburg provincial authorities.

Four intervention schools participated in HPSF and started implementing HP changes in November 2015. Since the schools themselves decided on the adoption and implementation of HP changes, some differences existed between them. School 1 (S1) and School 2 (S2), referred to as the ‘full HPSF’, decided to implement the two top-down changes, i.e., the lunch and the structured PA and cultural sessions [28]. To realize these changes during the lunch break, both schools extended the lunch break period by about 60 min. Therefore, children attended school to approximately 15:30/15:45 instead of 15:00. Both schools also implemented several additional HP changes, that is, they both provided water bottles to all children, improved their school’s health policy, and started with an educational lunch. The two schools implemented all HP changes in a comparable way and had similar support from external partners [28]. School 3 (S3) and School 4 (S4), referred to as the ‘partial HPSF’, decided to only implement the structured PA and cultural sessions each day. They did not provide a healthy lunch nor did they increase their lunch break time or implement additional HP changes [28]. The effects of the full and partial HPSF after a one- and two-year follow-up were investigated in two previous studies [32,33]. Significant favorable intervention effects after one- and two-years’ follow-up were found for the full HPSF on children’s dietary behaviors for, among others, school water consumption and lunch intake of vegetables and dairy products. Children’s sedentary time and light PA significantly improved after two years’ follow-up. Almost no significant favorable results on children’s health behaviors were found in the partial HPSF. In addition, results have shown that children’s BMI z-scores in both the full and the partial HPSF significantly decreased after two years’ follow-up. This favorable effect was already significant after one year’s follow-up in the partial HPSF, but not in the full HPSF.

### 2.2. Study Design

The current study was part of an overall study investigating HPSF, which included the four intervention schools and four control schools [30]. All the schools are members of the regional educational board ‘Movare’ situated in the Parkstad region in the southern part of the Netherlands. This region is characterized by a low average socioeconomic status, and unhealthy behaviors and overweight are highly prevalent compared to the rest of the Netherlands [34,35]. Ethical approval (14-N-142) for the overall study was given by the Medical Ethics Committee Zuyderland, located in Heerlen (Parkstad, the Netherlands). The current study incorporated two different study designs which were previously used in the overall study: (1) A longitudinal quasi-experimental study design to investigate the effects in each school [32,33], and (2) a mixed-methods study design to assess the four schools’ context and its moderating role [28]. Measurements for the quasi-experimental study were conducted during one week of measurements from September–November of 2015 (T0), 2016 (T1), and 2017 (T2). All children (aged 4 to 12) and their parents (*n* = 2326 at T0) from the eight schools were invited to participate in the study. All the participants were required to complete an informed consent form, signed by (both) parents. In the mixed-methods study, a contextual action-oriented research approach (CARA) was used as in Reference [31], which focused on contextual differences and the use of monitoring and inducing feedback loops to support and evaluate the processes of change. Data were collected in the four intervention schools over three years (2014–2017), that is, the development year (2014–2015) and the first two years of implementation (2015–2017) of the HPSF. The overall consent of the schools’ employees (school coordinator, teachers, external pedagogical employees) was obtained by consent of the director of the school.

### 2.3. Measures

#### 2.3.1. Effect Measures

Children’s BMI was assessed using anthropometric measurements of height and weight during physical education lessons. BMI z-scores were calculated using Dutch reference values as in Reference [36]. Children’s PA behaviors were assessed using accelerometry (Actigraph GT3X+, ActiGraph, Pensacola, FL, US, 30Hz, 10s epoch). The activity levels, in counts-per-minute (CPM), were classified using Evenson’s cut-off points [37]: sedentary behavior (SB; ≤100 CPM), light PA (LPA; 101–2295 CPM), and moderate-to-vigorous PA (MVPA; ≥2296 CPM). Children’s dietary behaviors were assessed through questionnaires addressed to the parents and children. The measures of the parents’ questionnaire were combined into two total scores (mean days/week): one for healthy dietary behaviors (breakfast, fruits, vegetables, and water), and one for unhealthy dietary behaviors (sugar-sweetened beverages and snacks). Two children’s questionnaires were used to obtain information about the children’s school water consumption and their lunch intake. The intake of specific food types (grains, butter, dairy, fruits, vegetables, and water) were summed, and a dichotomous variable was created to study whether the children consumed at least two of the food types during lunch. A more detailed description of the data collection procedures and the specific effect measures has been reported in the effect evaluation studies [32,33].

#### 2.3.2. Context Measures

##### Characteristics of the School Population

The number of children and teachers in each school was obtained, and their demographics and starting situation regarding health and health behaviors were assessed. The number of children in each school, their gender, ethnicity, and study year at baseline were collected from the database of the educational board Movare. Children’s ethnicity was determined by the country of birth of both parents and divided into: (1) Western background (including the Netherlands) and (2) non-Western background as described in Reference [38]. If one or both parents was born in a non-Western country, the child’s ethnicity was assigned to non-Western. A digital questionnaire for the parents was used to obtain information about the children’s socioeconomic status (SES), which was calculated as the mean of standardized scores on maternal education level, paternal educational level, and household income (adjusted for household size) [39]. The mean scores were categorized into low, middle, and high SES scores based on tertiles. To examine the children’s starting situation on health and health behaviors, the mean baseline scores per school of the above-mentioned effect measures were used. The number of teachers in each school was obtained from the school coordinators. Demographics and the starting situation of the teachers were collected from the teachers themselves by including additional questions on the HP practices questionnaire (see next section) regarding their gender, date of birth, the number of years employed by the school, and their height and weight. The latter was used to calculate their BMI.

##### HP Practices of the Teachers

A paper-based questionnaire was used to gain insight into the nutrition-related and PA-related HP practices of teachers at school, e.g., modelling behavior and involving children in nutrition or PA-related activities. The questionnaire was based on previous work by Gevers et al. [40,41] and O’Connor et al. [42], in which acceptable to good test-retest reliability of their instruments was found. The questionnaire was filled out annually by teachers at the beginning of the school year and consisted of 30 items (13 nutrition-related practices and 17 PA-related practices). Each item described a practice using a statement, followed by some examples. Participants responded on a Likert scale from 1 (completely disagree) to 5 (completely agree).

##### Perceived Barriers to the Implementation of HP Changes

To gain insight into the perceived barriers to the implementation of HP changes, a 46-item questionnaire was used. The questionnaire was distributed by e-mail and all implementers, i.e., teachers and external pedagogical employees, were asked to complete it digitally or in writing. The questionnaire was completed twice a year; for the current study, we included the data obtained prior to the start of the HPSF (T0) and after two years of implementation (T2). The questionnaire was based on the Measurement Instrument for Determinants of Innovations (MIDI), a Dutch questionnaire developed by Fleuren et al. [43]. They developed it through a systematic review of empirical studies and a Delphi study amongst implementation experts. The questionnaire has been used in many different implementation studies, especially in the school setting, although no specific research has been conducted to evaluate its validity and reliability [29]. Items were formulated as a statement, and responses to each statement ranged from 1 (totally disagree) to 10 (totally agree). The items were related to possible barriers regarding: (a) the users, i.e., the implementers themselves (*n* = 13); (b) the innovation, i.e., the HP changes (*n* = 7); (c) the support (*n* = 9); (d) the organization, i.e., the school (*n* = 13); and (e) the socio-political environment (*n* = 4). For each category, a mean score was calculated (maximum two missing).

##### HP Elements in the School

In the HPSF research, we used the term HP elements for initiatives in the school that potentially add to school-wide health promotion. A short questionnaire was filled out in the four intervention schools to gain insight into all these HP elements. Prior to the start of the HPSF (T0), it was done by interviewing the HPSF school coordinator, and after two years (T2), the school coordinators filled out the questionnaire themselves. The HP elements were divided into four themes: school routine, policy, education, and the environment. Elements regarding school routine (*n* = 7) were determined using questions on the use of energizers, drinking water during classes, the lunch in school, PA after lunch break, PA after school, the existence of working groups, and the involvement of parents. Elements regarding policy (*n* = 7) were determined using questions on rules and policy on snacks, lunch, treats, sugar-sweetened beverages, sport and energy drinks, water, and special policy on school events. Elements regarding education (*n* = 7) were determined using questions on having an educational lunch, swimming lessons, the number of minutes per week of physical education classes, and the use of four specific classroom-based programs regarding a healthy lifestyle. Elements regarding environment (*n* = 7) were determined using questions on the presence of a school vegetable garden, a bicycle parking area, a sports hall in the neighborhood, the use of volunteers to help children to cross a busy road, having a safe route to school, having an active schoolyard, and whether the schoolyard was open after school hours. The results were combined and translated into an overall score for that theme to indicate the extent to which it was present in the school (absent (-), minimally present (X), moderately present (XX), or largely present (XXX)).

##### Dominating Organizational Issues

A dominating organizational issue can be anything that could distract a school’s focus from its regular work and the implementation of the HPSF, e.g., staff turnover. Insight into existing dominating organizational issue(s) in the four schools was gained using several methods. The annual interviews with the HPSF school coordinators provided insight, and open questions were added to the barrier questionnaire, e.g., ‘Do other issues in school exist that influence the implementation of the Healthy Primary School of the Future?’ Furthermore, minutes of HPSF meetings that were held on an overarching level or on a school level, as well as formal and informal talks with people in the schools, provided insight into any existing dominating organizational issue. The dominating organizational issues were indicated per school as absent (-) or present (X).

### 2.4. Analyses

The analyses were conducted in five steps to investigate the four school contexts, the effects of HPSF in each school, and whether aspects in the context related to larger favorable effects.

Step 1: Assessing the four school contexts

Descriptives were used for the quantitative context measures. The qualitative context measures were described based on whether they were present in each school context and to what extent. The specific context in each school was assessed by comparison with the other schools.

Step 2: Comparing the school contexts between the schools with similar HP changes

Aspects in the school context were compared between the schools with similar HP changes. This meant that we compared the context of the full HPSF schools, i.e., S1 versus S2, and the context of the partial HPSF schools, i.e., S3 versus S4. Major differences between the contexts were described.

Step 3: Assessing the effects of HPSF in each school

IBM SPSS Statistics for Windows (version 23.0, IBM Corp, Armonk, NY, USA) was used to analyze the effects of HPSF in each school. Linear mixed-model analyses were conducted for the continuous effect measures and generalized estimating equations for the binary effect measures. These analyses, as well as the imputation method to handle missing data, were similar to the studies in which the effects of the full and partial HPSF on children’s BMI z-score and health behaviors were investigated [32,33]. A two-sided p-value ≤0.05 was considered statistically significant. Standardized effect sizes (ES) were determined for continuous effect measures, which were computed as the pooled estimated mean difference divided by the square root of the pooled residual variance at baseline. Odds ratios (OR) were determined for the binary effect measures.

Step 4: Comparing the effects of HPSF between the schools with similar HP changes

The effect sizes/odds ratios of all effect measures derived from Step 3 were compared between S1 and S2, and between S3 and S4. Similarities and differences were described.

Step 5: Exploring whether aspects in the school context relate to larger favorable effects of HPSF

This step was based on the principles of qualitative comparison analysis (QCA) [44,45]. QCA is a case-oriented approach that examines which aspects, alone or in combination with other aspects, are necessary or enough to produce an outcome. Using the principles of QCA, we aimed to explore whether aspects in the context relate to the larger favorable effects of HPSF. The findings from Step 2 and 4 were combined to conduct this step.

## 3. Results

The results were described according to the five steps of analysis. Table 1 presents all the results.


*Step 1: Assessing the four school contexts*


School Population

S1 was characterized by a school team with the highest number of teachers, as well as having the highest mean age of teachers. The children in this school had the most favorable starting situation regarding the BMI z-score, i.e., lowest z-scores, as well as PA behaviors, i.e., most time spent in PA. However, regarding children’s dietary behaviors, they had the least favorable starting situation. S2 was characterized by the least favorable starting situation of teachers, that is, their self-reported BMI was the highest compared to the teachers in the other schools. In S2 and S3, the school population, both children and teachers, was smallest at the start of the HPSF and it included the highest percentage of children who were part of the low SES tertile. Furthermore, S3 was characterized by the highest percentage of children with a Western ethnicity and the most favorable starting situation of children regarding their dietary behaviors. The school team in S3 consisted completely of female teachers, and they had been employed in their school for the shortest amount of time compared to the teachers in the other three schools. S4 was characterized by the highest number of children, as well as having the lowest mean age of children.

HP practices of teachers

S1 had the most and largest improvements in teachers PA-related practices. S2 had the most and greatest improvements in the teachers’ nutrition-related practices, and they were also the most favorable at T2 compared to the other schools. In S3, the teachers’ PA-related practices at the start of HPSF were the most favorable and many practices remained the most favorable at T2. In S4, the nutrition-related practices were the least favorable at both T0 and T2.

Perceived barriers to the implementation of HP changes

Teachers in S1 perceived the most barriers to implementation at both T0 and T2. External pedagogical employees in S1 perceived the least barriers to implementation of the HP changes. The opposite was observed in S3, where teachers perceived the least barriers and external pedagogical employees the most barriers. More detailed results, i.e., the results on each specific barrier in each school, were reported in Bartelink et al. [28].

HP elements in school

S1, S2, and S4 had limited HP elements at T0. S1 and S2 had improved greatly in all aspects at T2: policy, education, the environment, and school routine; whilst the improvements in S4 were limited. In S3, several HP elements already existed at the start of the HPSF. They did not show much improvement at T2. More detailed information on the specific HP elements in each school was reported in Bartelink et al. [28].

Dominating organizational issues

S1, S2, and S3 had to deal with a dominating organizational issue. S1 arose from the merger of two separate schools at the start of the HPSF, as well as having moved to a new school building. This merger process created a new way of working in the school. S2 had to deal with a merger as well. This merger was realized in September 2016, after the first year of the HPSF. For this merger, the school building had to be renovated, so they had to move to a temporary location with limited PA possibilities in and around the school, for the first year of the HPSF. This temporary location limited the implementation of HPSF. S3 had to deal with a major staff turnover at the start of the HPSF. This turnover had contributed to the decision of the school not to provide a healthy lunch.


*Step 2: Comparing the school contexts between the schools with similar HP changes*


The full HPSF: S1 versus S2

Differences were observed in four of the five contextual aspects.

School population: A larger percentage of children in S2 were part of the lowest SES tertile (32.3%), compared to S1 (24.0%). Children’s starting positions differed as well: the children in S1 had a more favorable mean BMI z-score (S1: mean (standard deviation (SD)) 0.028 (1.00) versus S2: 0.092 (1.02)) and they were more physically active (e.g., light PA in S1: (mean % per day (SD)) 32.8% (5.48) and in S2: 30.6% (5.98)). Dietary behaviors (in school) were more favorable in S2 (e.g., healthy behaviors in S1: mean (SD) 5.06 (1.16) and in S2: 5.33 (1.05); minimum of two food types during lunch in S1: 78.7% and in S2: 84.6%).

HP practices of teachers: The greatest and most improvements in PA-related teacher practices were found in S1 (e.g., encouragement in S1: T0 (mean (SD)) 4.2 (0.71), T2 4.6 (0.57)) and in S2 (T0 4.5 (0.64), T2 4.4 (0.61)), and in nutrition-related practices in S2 (e.g., healthy modelling in S2: T0 4.3 (1.10), T2 4.7 (0.56)).

Perceived barriers to the implementation of HP changes: The external pedagogical employees in S1 perceived the least barriers at T0 and T2, but in S2, the greatest improvements in perceived barriers could be observed over the two years (e.g., innovation-related barriers in S1: T0 (mean (SD)) 7.6 (1.42), T2 7.6 (0.60), and in S2: T0 6.9 (0.20), T2 7.3 (1.04)).

Dominating organizational issues*:* Even though both S1 and S2 had to deal with a merger process, the impact was different, i.e., in S1, it reinforced the implementation of HPSF, whilst in S2, it limited the implementation.

HP elements in school: No differences were observed in this contextual aspect.

The partial HPSF: S3 versus S4

Differences in context were observed in three of the five contextual aspects.

School population: A larger percentage of children in S3 (38.4%) were part of the lowest SES tertile compared to S4 (28.1%). The starting situation of the children was also different: more favorable in S3 regarding BMI z-score (S3: 0.082 (1.01) vs. S4: 0.099 (0.91)) and dietary behaviors (e.g., school water consumption in S3: (mean (SD)) 2.94 (1.22) and in S4: 1.93 (1.06)), and more favorable in S4 regarding PA behaviors (e.g., sedentary time in S3: (mean % per day (SD)) 61.6% (6.54) and in S4: 60.3% (7.29)). Furthermore, teachers in S3 had been employed in their school for a shorter amount of time (mean (SD): 0.92 (1.24)) compared to those in S4 (12.64 (8.56)).

HP practices of teachers: Teacher’s PA practices at the start of the HPSF were more favorable in S3 than in S4 (e.g., involving children in PA activities was in S3: 4.5 (0.53) and in S4: 3.5 (1.03)).

Perceived barriers to the implementation of HP changes: A larger decline in perceived barriers of teachers was observed in S3 compared to S4 over the two years (e.g., support-related barriers in S3: T0 7.1 (1.80), T2 8.6 (0.65) and in S4: T0 7.5 (0.87), T2 8.0 (0.94)). External pedagogical employees of S4 perceived fewer barriers at both time points compared to S3 (e.g., user-related barriers in S3: T0 7.5 (0.68), T2 7.2 (0.55) and in S4: T0 7.6 (0.65), T2 7.9 (0.38)).


*Step 3: Assessing the effects of HPSF in each school*


The largest effects on children’s BMI z-score were found in S1 (ES = −0.11) and the smallest in S3 (ES = −0.04) (Table 1; Appendix A). For the effects on children’s PA behaviors, i.e., the time children spent sedentary, in light PA and in MVPA, the largest effects were found in S1 and the smallest in S3. The effect in S4 on the time children spent in MVPA was comparable to S1 (ES = 0.15). Overall dietary behaviors improved most in S1, i.e., an increase in healthy dietary behaviors (ES = 0.25) and a decrease in unhealthy dietary behaviors (ES = −0.13). The least favorable effect on healthy dietary behaviors, with a negative effect size, was found in S4 (ES = −0.08). Regarding unhealthy dietary behaviors, an adverse effect was also found, that is, in S3, the ES was 0.06. The largest effects were found in S2 on school dietary behaviors, i.e., school water consumption (ES = 1.17) and the intake of at least two healthy food types during lunch at school (OR = 3.96). The least favorable, and even adverse, effects were found in S3 (school water consumption: ES = −0.20; lunch intake: OR = 0.20).


*Step 4: Comparing the effects of HPSF between the schools with similar HP changes*


The full HPSF: S1 versus S2

Larger favorable effects were found in S1 compared to S2 for the children’s BMI z-score, their PA behaviors, and their overall dietary behaviors. Looking at the effects on dietary behaviors in school, the effects were similar or more favorable in S2.

The partial HPSF: S3 versus S4

The favorable effects on all outcome measures were larger in S4 compared to S3, except for overall healthy dietary behaviors.


*Step 5: Exploring whether aspects in the school context relate to larger favorable effects of HPSF*


Five aspects in the context appeared to be related to larger favorable effects. Larger effects were found in schools with: (1) fewer children in the lowest SES tertile; (2) more favorable starting positions of children regarding their health behaviors; (3) most improvements in nutrition and/or PA-related practices of teachers, specifically related to modelling and encouragement; (4) least barriers perceived by the external pedagogical employees; and (5) in the schools that used the opportunity created by a dominating organizational issue, e.g., merger process, to synergize it with the implementation of the HPSF.

## 4. Discussion

The current study assessed and compared the contexts and effects of HPSF in four schools and explored whether aspects in the context related to larger favorable effects. The results showed that the four school contexts were different at the start of the HPSF, and that they evolved differently during the two years of HPSF, and that the effects of HPSF were different for the four schools. These findings underline our rationale that the school context influences the effects of school health promotion efforts. These varying effects across schools can be seen as the result of the nonlinearity of the system and the interaction of contextual aspects with the HP changes in the school. Since each complex adaptive school system is unique and can react in a different way, varying effects can be expected and they seem to represent the natural variation within complex adaptive systems [46].

The findings in the current study showed that potentially moderating contextual aspects were found on the level of the children, the employees, and the school itself. These levels were not separated in a complex adaptive school system, but they also continuously interacted with each other. It is challenging and maybe even impossible to fully understand this complex system behavior and its impact on the effects on children’s health and health behaviors. In this study, however, we aimed to take a first step towards this understanding by exploring the contextual aspects in the schools that appeared to be related to larger effects.

Concerning the level of the children, we found that larger effects were observed in the schools in which a smaller percentage of children were part of the lowest SES group, and in which children had the most favorable starting position regarding their health behaviors. Even though, on average, the effects were favorable for all schools and the focus was already on a low SES area, the findings regarding SES seemed to indicate that HPSF is more favorable for the higher SES groups. This suggests that HPSF might still contribute to the socioeconomic health inequity gap [47]. The moderation effect of SES indicates that the intervention outcomes interact with the children’s background in the home context. It underlines that the school system is an open system, and that effects of HP changes in the school can also be moderated by aspects in the home context or neighborhood [48,49]. Moreover, the findings showed larger effects of HPSF, not only in the higher SES schools, but also in the schools in which children had the most favorable starting position regarding their health behaviors. The association between these two aspects was investigated in-depth for the children in the HPSF schools by Vermeiren et al. [50], and they were in line with other studies which showed that less favorable health behaviors tended to be associated with a lower SES [51]. This means that a school that includes more children with a lower SES background may also have more children with less favorable health behaviors, and vice versa. This seems to indicate that the moderation of these two child characteristics is clustered. Overall, the findings suggested that even though HPSF was beneficial in all schools, it may lead to smaller effects in the schools that included the most disadvantaged group of children. It should be examined whether further adaptation of the HPSF to the school’s population is needed or whether these schools just need more time to achieve more favorable effects.

Concerning the level of the employees or intermediaries, we found larger effects in the schools with the most improvements in HP practices of the teachers (specifically related to modelling and encouragement of healthy nutrition and PA), and schools with the least barriers perceived by the external pedagogical employees. The moderating role of teachers’ HP practices was in line with the study by Gubbels et al., who investigated this in the childcare setting. They showed the importance of favorable food practices of employees at the childcare organizations, such as modelling behavior, for a healthy food intake by the children [52]. The findings of the current study suggested that by improving the HP practices of teachers, the larger effects of HP changes in the school could be achieved. Thus, it is recommended to directly intervene in these HP practices of teachers. The findings also showed that a focus on the perception of external pedagogical employees, the main implementers of the HP changes, can optimize the effects. These external employees were provided by childcare organizations and were employed to avoid increasing the teachers’ workload even further. This integration of the childcare organization during school hours was not intended to provide a temporary solution, but to provide professional employees for the implementation of the HP changes, and to change the school’s organization in a sustainable way. The findings in this study imply that to achieve larger effects, it is recommended to monitor the perception of these main implementers regularly to provide input for feedback loops. These feedback loops, also visualized in the program theory (Figure 1), should make it possible to understand and tackle perceived barriers. Overall, these findings on the level of the employees suggest that directly investing and intervening in them by improving teachers’ HP practices and monitoring, as well as tackling the barriers perceived by the main implementers, may contribute to achieving larger effects of a health promoting school initiative.

Regarding the level of the school itself, we found that larger effects were observed in schools when they were able to synergize existing organizational issues in the school with the HPSF. Some schools had to deal with a dominating organizational issue, e.g., merger process, that disrupted their normal functioning. Such a disrupting event in the school can create an opportunity for HPSF, i.e., the merger process can build up momentum for a new start, which helps to create a new way of working in which HPSF is also included. Therefore, it is recommended to gain insight into whether organizational issues exist in the school and how this can be used as an opportunity to build up momentum for HPSF.

The findings in this study demonstrate that a broad insight into the school context is crucial for understanding the intervention effects of HP changes in a complex adaptive school system. The focus should not merely be on intervention evaluation, but also on the context evaluation [12,18,53]. The findings suggest that it may result in an over- or under-estimation of the effects of HP changes when different school contexts are combined in the analyses. Therefore, it is recommended to also examine the effects separately for each school context. Moreover, the results of this study suggest that the average effect sizes of intervention outcomes do not provide a full answer regarding the effectiveness [46,54]. Larger effects may be achieved due to the interaction with specific contextual aspects, e.g., more children in the school with a higher socioeconomic background or organizational issues in the school. Therefore, when evaluating the effectiveness of HP changes, the focus should not only be on the effect sizes and outcomes, but also on aspects in the context that interacted with the HP changes. This context-oriented evaluation of HP changes contributes to a better understanding of the moderating role of the school context on the effects of HP initiatives. It may explain the variation in effects across schools, and it can provide insight on which contextual aspects to focus on or intervene in to optimize the effects.

### Strengths and Limitations

Several strengths and limitations of the study should be considered. Since HPSF was quite comparable between S1 and S2, and between S3 and S4, we saw an opportunity to explore the moderating role of the school context on the intervention effects of the HPSF. However, a limitation is that we could not determine whether differences in the effects between schools were due to differences in the implementation of the HPSF [9,11,13]. Furthermore, two comparisons are still limited; however, we were able to combine the results of both comparisons and form stronger conclusions about the moderating role of the school context. Future research should investigate whether the findings of this study also apply to other schools. Finally, assessing the school contexts had several limitations. We examined many of the contextual aspects in a quantitative manner, which may not fully capture each aspect. It was also impossible to fully assess and understand all aspects of each school’s context citing limitations in resources, time, and participant burden [21]. This might have led to missing important, possibly moderating, contextual aspects. Nevertheless, we were able to examine contextual aspects on all levels in the school and to focus on the aspects suggested by other researchers as relevant for improving school health promotion [8,12,26,29].

## 5. Conclusions

Similar HP changes lead to different outcomes across schools due to differences in the school context. Potentially moderating contextual aspects in the Healthy Primary School of the Future were found at the level of the children, the employees, and the school itself. When evaluating the effectiveness of HP changes, the focus should not only be on the overall effect sizes, but also on which aspects in the context interacted with the HP changes. The adoption of a complex adaptive systems perspective contributes to a better understanding of the variation in effects across schools and it can provide insight on which contextual aspects to focus on or intervene in to optimize the effects of HP initiatives.

## Figures and Tables

**Figure 1 ijerph-16-02432-f001:**
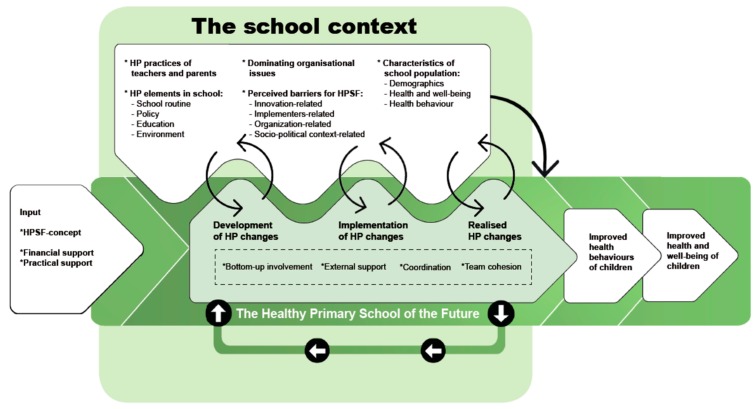
Program theory derived from Bartelink et al. [28]. The left side of the model shows the input, which is an ‘event’ that attempts to positively disrupt the pre-existing dynamics in the school context to integrate health promotion. After the introduction of HPSF into the school context, the process of development, implementation, and integration of HP changes develops in the school. During this process, it is hypothesized that HPSF will continuously interact with the school context. The loop in the bottom of the program theory visualizes the key assumption that realized changes may shift the school’s norms toward a focus on health and well-being, thereby creating momentum for additional HP changes. Overall, the process of change should lead to the realization of HP changes that fit the school’s context. The combination and interaction of all these contextualized HP changes should impact children’s health behaviors and, through this, their health and well-being. A key assumption in the cause–effect relation concerns non-linearity: it is assumed that small changes in a school can produce large effects at a so-called ‘tipping’ point. The arrow in the top right of the model visualizes the moderating role of the school context. The key assumption is that even when a change is similar, the school context will determine its impact.

**Table 1 ijerph-16-02432-t001:** The effects of HPSF in each school and the four school contexts.

**Effects of HPSF in Each School (Effect Sizes) ***								
**Full HPSF**				**Outcome measures**	**Partial HPSF**			
**School 1**		**School 2**			**School 3**		**School 4**	
**T0–T2**		**T0–T2**			**T0–T2**		**T0–T2**	
−0.11		−0.05		**BMI z−score**	−0.04		−0.08	
−0.27		−0.08		**Sedentary time**	−0.03		−0.08	
0.26		0.09		**Light PA**	0.01		0.02	
0.15		0.05		**MVPA**	0.04		0.15	
0.25		0.03		**Healthy dietary behaviors**	0.08		−0.08	
−0.13		−0.07		**Unhealthy dietary behaviors**	0.06		−0.02	
1.16		1.17		**Water consumption in school**	−0.20		0.14	
2.52		3.96		**Minimum of two food types during lunch** (OR)	0.20		0.94	
**The Four School Contexts**								
**Characteristics of the school population—Children**								
**Full HPSF**				**Contextual aspects**	**Partial HPSF**			
**School 1**		**School 2**			**School 3**		**School 4**	
**T0**		**T0**			**T0**		**T0**	
324		234		**Number of children** (*N*)	233		389	
48.6		51.0		**Gender** (% boys)	45.7		49.5	
8.04 (2.21)		8.55 (2.22)		**Age in years** (mean (SD))	8.45 (2.34)		7.82 (2.43)	
24.0		32.3		**SES** (% lowest tertile)	38.4		28.1	
94.8		94.3		**Ethnicity** (% western)	98.4		94.9	
**Characteristics of the school population—Teachers**								
**Full HPSF**				**Contextual aspects**	**Partial HPSF**			
**School 1**		**School 2**			**School 3**		**School 4**	
**T0**		**T0**			**T0**		**T0**	
26		15		**Number of teachers** (N)	16		21	
19.2		33.3		**Gender** (% men)	0		22.7	
45.35 (13.71)		44.60 (11.81)		**Age in years** (mean (SD))	41.80 (13.46)		41.27 (10.74)	
12.31 (12.72)		11.47 (7.28)		**Working in the school in years** (mean (SD))	0.92 (1.24)		12.64 (8.56)	
**Starting situation of the school population—Children**								
**Full HPSF**				**Contextual aspects**	**Partial HPSF**			
**School 1**		**School 2**			**School 3**		**School 4**	
**T0**		**T0**			**T0**		**T0**	
0.028 (1.00)		0.092 (1.02)		**BMI z-score** (mean (SD))	0.082 (1.01)		0.099 (0.91)	
58.5 (6.76)		61.9 (7.45)		**Sedentary time** (% per day)	61.6 (6.54)		60.3 (7.29)	
32.8 (5.48)		30.6 (5.98)		**Light PA** (% per day)	31.1 (5.35)		32.4 (5.80)	
8.7 (2.63)		7.5 (2.72)		**MVPA** (% per day)	7.3 (2.58)		7.3 (2.51)	
5.06 (1.16)		5.33 (1.05)		**Healthy dietary behavior in days/week** (mean (SD))	5.24 (0.81)		5.07 (1.12)	
1.12 (0.64)		1.08 (0.66)		**Unhealthy dietary behavior in days/week** (mean (SD))	1.02 (0.53)		1.11 (0.63)	
2.16 (1.11)		2.20 (1.07)		**School water consumption: range 0-3** (mean (SD))	2.94 (1.22)		1.93 (1.06)	
78.7		84.6		**Minimum of two food types during lunch** (% yes)	94.7		79.3	
**Starting situation of the school population—Teachers**								
**Full HPSF**				**Contextual aspects**	**Partial HPSF**			
**School 1**		**School 2**			**School 3**		**School 4**	
**T0**		**T0**			**T0**		**T0**	
23.79 (3.16)		27.50 (4.61)		**Self-reported BMI in kg/m^2^ (mean (SD))**	23.23 (2.67)		24.23 (3.00)	
**Nutrition-related teacher practices** (range 1-5; presented as mean (SD)) **								
**Full HPSF**				**Contextual aspects**	**Partial HPSF**			
**School 1**		**School 2**			**School 3**		**School 4**	
**T0**	**T2**	**T0**	**T2**		**T0**	**T2**	**T0**	**T2**
4.4 (±0.80)	4.5 (±0.74)	4.3 (±1.10)	4.7 (±0.56) **Δ**	**Healthy modelling**	4.2 (±0.97)	4.1 (±0.99) **Δ**	4.1 (±1.05)	4.0 (±0.89) **Δ**
4.0 (±1.20)	4.6 (±0.69)	4.2 (±0.94)	4.9 (±0.62) **Δ**	**Prevent unhealthy modelling**	4.1 (±1.10)	3.5 (±1.20)	3.5 (±1.18)	3.7 (±1.17) **Δ**
4.5 (±0.51)	4.6 (±0.56)	4.3 (±0.88)	4.9 (±0.46) **Δ**	**Encouragement**	4.7 (±0.48)	4.5 (±0.54) **Δ**	4.5 (±0.60)	4.3 (±0.75) **Δ**
4.4 (±0.63)	4.5 (±0.63)	4.1 (±1.16)	4.5 (±0.51) **Δ**	**Involving**	4.3 (±0.95)	4.4 (±0.52) **Δ**	4.3 (±1.03)	4.1 (±0.95)
4.4 (±0.75)	4.4 (±0.57)	4.3 (±0.72)	4.7 (±0.45) **Δ**	**Discussing**	4.5 (±0.53)	4.3 (±0.71)	4.2 (±0.61)	4.1 (±0.68) **Δ**
4.2 (±0.77)	4.5 (±0.69) **Δ**	4.2 (±0.68)	4.5 (±0.61) **Δ**	**Educating**	4.6 (±0.70)	4.1 (±0.64)	4.1 (±0.68)	4.3 (±0.74) **Δ**
4.0 (±0.66)	4.2 (±0.69)	3.9 (±0.80)	4.3 (±0.56) **Δ**	**Providing feedback**	3.7 (±1.06)	4.1 (±0.35) **Δ**	4.1 (±0.99)	3.8 (±1.02)
3.7 (±1.23)	3.7 (±1.31)	3.6 (±1.24)	4.1 (±0.71) **Δ**	**Visibility**	4.0 (±0.82)	3.0 (±1.07)	3.7 (±1.56)	3.7 (±0.96 **Δ**
4.4 (±0.85)	4.5 (±0.51)	4.3 (±0.98)	4.7 (±0.48) **Δ**	**Routines**	4.2 (±0.83)	4.4 (±0.52) **Δ**	4.2 (±0.81)	4.2 (±0.82)
3.5 (±1.07)	4.5 (±1.30) **Δ**	3.5 (±1.13)	3.8 (±1.03)	**Monitoring**	3.5 (±1.43)	3.8 (±0.46)	2.9 (±1.41)	3.4 (±1.59) **Δ**
4.5 (±0.91)	4.9 (±0.65) **Δ**	4.5 (±0.52)	4.7 (±0.58)	**Rules**	4.4 (±0.70)	4.3 (±0.89)	4.2 (±0.96)	4.4 (±0.89) **Δ**
3.3 (±1.09)	3.6 (±1.15) **Δ**	3.6 (±0.91)	3.9 (±1.73) **Δ**	**Pressure to eat *****	3.5 (±1.35)	3.0 (±0.76) **Δ**	3.6 (±1.37)	3.2 (±1.47)
2.5 (±1.75)	2.4 (±1.78)	2.8 (±1.52)	1.5 (±1.17) **Δ**	**Instrumental feeding *****	2.8 (±1.48)	2.1 (±1.25) **Δ**	2.9 (±1.39)	2.8 (±1.63)
**PA-related teacher practices** (range 1–5; presented as mean (SD)) **								
**Full HPSF**				**Contextual aspects**	**Partial HPSF**			
**School 1**		**School 2**			**School 3**		**School 4**	
**T0**	**T2**	**T0**	**T2**		**T0**	**T2**	**T0**	**T2**
4.2 (±0.71)	4.6 (±0.57) **Δ**	4.5 (±0.64)	4.4 (±0.61)	**Encouragement**	4.5 (±0.71)	4.6 (±0.52)	4.2 (±0.81)	4.5 (±0.78) **Δ**
4.1 (±0.89)	4.6 (±0.83) **Δ**	4.1 (±0.83)	4.1 (±0.71)	**Rewarding**	4.1 (±0.99)	4.3 (±0.89) **Δ**	4.1 (±1.11)	4.0 (±1.00)
3.8 (±0.91)	4.1 (±0.95) **Δ**	4.2 (±0.68)	3.9 (±0.71)	**Involving**	4.5 (±0.53)	4.5 (±0.54)	3.5 (±1.03)	4.2 (±0.78) **Δ**
3.7 (±1.01)	4.0 (±0.98) **Δ**	3.9 (±0.92)	3.9 (±0.71)	**Healthy modelling**	3.9 (±0.88)	4.1 (±0.84)	3.4 (±1.37)	3.9 (±1.15) **Δ**
4.1 (±0.74)	4.2 (±0.77)	4.0 (±0.85)	4.2 (±0.50) **Δ**	**Discussing**	4.5 (±0.53)	4.3 (±0.71)	4.3 (±0.80)	4.3 (±0.76) **Δ**
4.2 (±0.63)	4.5 (±0.74) **Δ**	4.2 (±0.56)	4.2 (±0.69)	**Educating**	4.7 (±0.68)	4.6 (±0.74) **Δ**	4.5 (±0.80)	4.3 (±0.74)
3.9 (±0.94)	3.7 (±1.11) **Δ**	3.4 (±1.30)	3.8 (±0.83)	**Discouragement *****	3.9 (±1.10)	3.0 (±0.93) **Δ**	4.4 (±0.85)	3.9 (±1.06)
3.6 (±0.90)	4.2 (±0.97) **Δ**	3.7 (±1.10)	3.7 (±0.89)	**Providing feedback**	3.7 (±1.16)	4.0 (±0.76) **Δ**	3.5 (±1.37)	3.7 (±1.08)
4.4 (±1.03)	4.3 (±1.00)	4.1 (±0.64)	4.3 (±0.45) **Δ**	**Availability**	4.3 (±0.48)	4.3 (±0.89) **Δ**	4.1 (±1.07)	4.0 (±1.00)
3.6 (±1.50)	4.0 (±1.59) **Δ**	3.1 (±1.30)	3.5 (±0.90) **Δ**	**Visibility**	3.3 (±1.34)	4.0 (±1.69) **Δ**	3.6 (±1.71)	3.1 (±1.32)
4.2 (±0.71)	4.4 (±0.62) **Δ**	4.6 (±0.63)	4.6 (±0.51)	**Accessibility**	4.6 (±0.52)	4.6 (±0.52) **Δ**	4.5 (±0.67)	4.5 (±0.59) **Δ**
3.9 (±0.77)	4.5 (±1.18) **Δ**	4.1 (±0.80)	4.1 (±0.78)	**Routines**	4.3 (±0.82)	4.3 (±0.71)	3.8 (±1.15)	4.4 (±0.71) **Δ**
2.8 (±1.20)	3.2 (±1.54)	2.8 (±0.94)	2.7 (±1.45) **Δ**	**Warning PA *****	2.6 (±1.35)	2.4 (±1.30) **Δ**	3.1 (±1.68)	3.1 (±1.64)
3.5 (±1.11)	3.9 (±1.45) **Δ**	3.1 (±1.03)	3.2 (±1.17)	**Monitoring**	3.1 (±1.29)	3.0 (±0.93) **Δ**	3.3 (±1.67)	3.0 (±1.57)
4.2 (±0.98)	4.3 (±0.67) **Δ**	4.4 (±0.63)	4.4 (±0.61)	**Rules**	4.5 (±0.53)	4.6 (±0.74) **Δ**	4.5 (±1.14)	4.5 (±0.66)
4.3 (±0.62)	4.1 (±0.64) **Δ**	4.5 (±0.83)	4.4 (±0.50)	**Pressure to be physically active *****	4.5 (±0.53)	4.6 (±0.52) **Δ**	4.2 (±1.01)	4.5 (±0.78)
3.6 (±1.44)	4.0 (±1.24)	3.5 (±1.25)	3.4 (±1.30) **Δ**	**Instrumental feeding *****	4.4 (±1.08)	3.8 (±1.58) **Δ**	4.0 (±1.25)	4.0 (±1.37)
**Teacher—perceived barriers for HPSF, related to** (range 1–10; presented as mean (SD))								
**Full HPSF**				**Contextual aspects**	**Partial HPSF**			
**School 1**		**School 2**			**School 3**		**School 4**	
**T0**	**T2**	**T0**	**T2**		**T0**	**T2**	**T0**	**T2**
7.9 (±0.51)	7.6 (±0.75)	8.5 (±0.53)	8.4 (±0.72) **Δ**	**User**	8.1 (±0.96)	8.6 (±0.73) **Δ**	7.9 (±0.57)	8.2 (±0.76)
6.3 (±1.20)	6.8 (±1.28) **Δ**	7.3 (±0.64)	7.3 (±0.90)	**Innovation**	7.5 (±1.21)	8.3 (±0.97) **Δ**	7.4 (±0.83)	8.0 (±0.82)
6.8 (±0.93)	7.8 (±1.17) **Δ**	7.8 (±0.59)	7.7 (±0.76)	**Support**	7.1 (±1.80)	8.6 (±0.65) **Δ**	7.5 (±0.87)	8.0 (±0.94)
7.3 (±1.43)	7.3 (±0.97)	7.5 (±1.02)	7.7 (±0.82) **Δ**	**Organization**	7.7 (±1.56)	8.5 (±0.80) **Δ**	8.2 (±2.10)	8.1 (±0.87)
8.0 (±0.87)	7.6 (±0.75)	8.1 (±0.96)	8.3 (±0.83) **Δ**	**Socio-political context**	7.7 (±2.08)	8.8 (±0.95) **Δ**	7.8 (±1.02)	8.3 (±0.93)
**External pedagogical employee—perceived barriers for HPSF, related to** (range 1–10; presented as mean (SD))								
**Full HPSF**				**Contextual aspects**	**Partial HPSF**			
**School 1**		**School 2**			**School 3**		**School 4**	
**T0**	**T2**	**T0**	**T2**		**T0**	**T2**	**T0**	**T2**
8.2 (±0.94)	8.1 (±0.82)	8.0 (±0.71)	8.0 (±0.62) **Δ**	**User**	7.5 (±0.68)	7.2 (±0.55)	7.6 (±0.65)	7.9 (±0.38) **Δ**
7.6 (±1.42)	7.6 (±0.87)	6.9 (±0.20)	7.3 (±1.04) **Δ**	**Innovation**	6.8 (±0.67)	6.9 (±0.37)	6.9 (±0.73)	7.3 (±0.82) **Δ**
7.4 (±1.33)	8.1 (±0.60)	5.6 (±1.53)	7.8 (±0.62) **Δ**	**Support**	5.6 (±1.62)	6.6 (±1.30) **Δ**	7.0 (±1.10)	7.9 (±0.28)
8.0 (±1.01)	7.7 (±0.89)	6.3 (±1.68)	8.0 (±0.48) **Δ**	**Organization**	5.9 (±0.88)	6.9 (±0.53) **Δ**	7.4 (±0.40)	7.8 (±0.49)
8.5 (±1.01)	8.0 (±1.14)	8.3 (±0.35)	7.9 (±0.76) **Δ**	**Socio-political context**	6.1 (±0.72)	6.6 (±0.56) **Δ**	8.0 (±0.91)	7.3 (±0.90)
**Health-promoting elements in school** (absent (-), minimally present (X), moderately present (XX), largely present (XXX))								
**Full HPSF**				**Contextual aspects**	**Partial HPSF**			
**School 1**		**School 2**			**School 3**		**School 4**	
**T0**	**T2**	**T0**	**T2**		**T0**	**T2**	**T0**	**T2**
X	XXX	X	XXX	**Policy**	X	XX	-	X
-	XX	-	XX	**Education**	XX	XXX	X	XX
X	XX	X	XX	**Environment**	XX	XX	X	XX
X	XXX	X	XXX	**School routine**	XX	XX	XX	XX
**Dominating organizational issues** (absent (-) or present (X))								
**Full HPSF**				**Contextual aspects**	**Partial HPSF**			
**School 1**		**School 2**			**School 3**		**School 4**	
**T0–T2**		**T0–T2**			**T0–T2**		**T0–T2**	
X (Nov 2015)		X (Sep 2016)		**Merge**	-		-	
-		-		**Staff turn-over**	X (Sep 2015)		-	
-		X (school year 2015/2016)		**Temporary location**	-		-	
X (Nov 2015)		-		**New school building**	-		-	

* More information on the school-specific intervention effects of HPSF after the two years’ follow-up is included in Appendix A. ** More information on the specific nutrition and PA-related practices of teachers is included in Appendix A. *** A decrease is favorable for this practice. **Δ**
*=* The largest improvements between T0 and T2. Explanation of colors: The characteristics of the school population with the highest percentages or means were highlighted in blue; the most favorable scores at T0 and T2 were highlighted in green. When the schools had similar scores, both schools were highlighted. Abbreviations: BMI = body mass index; HPSF = the Healthy Primary School of the Future; MVPA = moderate to vigorous physical activity; OR = odds ratio; PA = physical activity; SD = standard deviation.

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
