# Peer review of "The Moderating Role of the School Context on the Effects of the Healthy Primary School of the Future"

_ijerph, 2019, doi:10.3390/ijerph16132432_

Round 1

Reviewer 1 Report

This is a very sharp and well-written article, which focuses on a key issue in health promotion. 

I have made a few comments to the authour:

- l 46: how would you argue that the HPS is a concept. In my view it is more of a strategy. I think this is an ill-choice of words. 

- l 59 - 61: the differentiation between the focus of your study and previous work is not clear. is it that you choose a particular setting?

- the HPS Framework is brought up in the introduction, then the argument moves onto health promotion in schools, which suggests a setting-based approach rather than references to the whole school Framework. Is this a standpoint? or does it need to be clarified in the text?

- figure 1 is very interesting, but the logic model is not clear. it would be easier for the reader to have some kind of guideline as to how to read the model (inputs are specified, but outcomes? outputs? processes?) Also this figure looks more like a logic model than a programme theory to me, as it is lacking explanations and references as to the leverage points activated by the project.

- how are the Momentum for traditionnal HP changes created? l99

- how did the team get authorization to collect data for BMI score? it considered as medical data, bound by secracy in my country. could the authors elaborate a little bit on this ethical issue?

- also the mechanism involved in the favorable change in BMI is not presented in the project's logic model. How can it be assumed that this effect results from implementation of the project?

- l 119-121: how long was the timeframe for the previous study? in other words how long did it take to observe changes in the BMI of children?

- from l 199, what do you mean by "element"?  is a rule on snacks a HP Policy which draws from a whole school approach? what about school Policy and how HP is included within curriculum development and school projects and so on.. 

- the paragraph starting l 313 is difficult to read. same comment with the next paragraph

- if I am not mistaken the argument l 382 to 383 (complex interactions) is one of the points made by the authors to justify the choice of questions and methods. it seems Strange to find it as a conclusion from the study

Overall, this article is a high quality description of the key issues in the field of health promotion reserach on implementation. The methods described are scientifically sound and the description is consistent with the expected validty criteria in research. The results make a key contribution to the field. I highly recomment this for publication. 

Author Response

Dear editor,

Thank you for allowing us to submit a revised version of our manuscript. We also want to thank the reviewers for their positive judgements and helpful comments on the manuscript. Below we explain how we incorporated the comments in our revision. In the manuscript, we have marked the changes in yellow.

Reviewer 1

This is a very sharp and well-written article, which focuses on a key issue in health promotion. I have made a few comments to the author:

- l 46: how would you argue that the HPS is a concept. In my view it is more of a strategy. I think this is an ill-choice of words. 

We agree with the reviewer that ‘strategy’ fits better. We have changed it in the manuscript.

Page 1, Line 46:However, even though this strategy to integrate health promotion into the school system is promising, … ‘.

- l 59 - 61: the differentiation between the focus of your study and previous work is not clear. is it that you choose a particular setting?

The difference between previous work and our study is that previous studies have mainly focused on the interaction between the HP changes and the context during implementation. Our focus is related to the influence of the context on the effects of HP changes, when implementation is comparable between schools. In other words, we focused on the moderating role of the context on the effects of HP changes. We argue that even when different schools implement HP changes in a similar way, the differences in context can result in different effects between schools. For example, both schools can implement educational lessons in a comparable way, but in one school it leads to large effects and in another school the effects are smaller. We argue that this can be due to a moderating role of the context. We visualized this in the programme theory by the moderator arrow in the top right of the model. We changed the text in the manuscript as follows:

Page 2, Line 59-63:Several studies have examined the role of the school context, but mainly focused on its interaction with HP changes during the implementation process (12, 25). The focus of this study was to examine the moderating role of the school context on the effects of HP changes when implementation is comparable between schools.’

- the HPS Framework is brought up in the introduction, then the argument moves onto health promotion in schools, which suggests a setting-based approach rather than references to the whole school Framework. Is this a standpoint? or does it need to be clarified in the text?

We thank the reviewer for this comment. We definitely did not want to make a standpoint. We meant to describe the specific HP changes as part of the whole system change. We changed the text as follows:

Page 1, Line 46-49: However, even though this strategy to integrate health promotion into the whole school system is promising, suboptimal results are often observed, due to, among other things, challenges regarding the implementation of specific health-promoting (HP) changes as part of this school-wide change and how to create meaningful impact (9-14).

Page 2, Line 55-58: Embracing this perspective of considering schools as complex adaptive systems means that it depends on the specific school context whether a specific HP change fits in a school, and that in each school the implementation process of a specific HP change is different (21, 22).

- figure 1 is very interesting, but the logic model is not clear. it would be easier for the reader to have some kind of guideline as to how to read the model (inputs are specified, but outcomes? outputs? processes?) Also this figure looks more like a logic model than a programme theory to me, as it is lacking explanations and references as to the leverage points activated by the project.

We agree with the reviewer that some guidance in reading the programme theory should be included. We added the information below as explanatory text to Figure 1 in the paper. In one of our previous papers, the process evaluation of HPSF, we discussed this programme theory in more detail [1]. In this process evaluation we made the choice to not define it as a logic model, as this refers to much to a causal process. To be in line with this process evaluation, we also refer to it as programme theory in the current paper.

Page 3, Line 87-94: The left side of the model shows the input, which can be seen as an ‘event’ that attempts to positively disrupt the pre-existing dynamics in the school context in order to integrate health promotion. After the introduction of HPSF into the school context, the process of development, implementation, and integration of HP changes develops in the school. During this process it is hypothesized that HPSF will continuously interact with the school context. The loop in the bottom of the programme theory visualizes the key assumption that realized changes may shift the school’s norms toward a focus on health and well-being, thereby creating momentum for additional HP changes. Overall, the process of change should lead to the realization of HP changes that fit the school’s context. The combination and interaction of all these contextualized HP changes should impact children’s health behaviours and, through this, their health and well-being. A key assumption in the cause-effect relation concerns non-linearity: it is assumed that small changes in a school can produce large effects at a so-called ‘tipping’ point. The arrow in the top right of the model visualizes the moderating role of the school context. The key assumption is that even when a change is similar, the school context will determine its impact.

- how are the Momentum for additional HP changes created? l99

The assumption is that the two top-down changes positively disrupts the way of working in the school system. This means that the school needs to find a new stability to be able to successfully integrate the two changes. This can create a shift in the system’s norms toward a focus on health and well-being, and during this instability in the school system, new needs, opportunities, and wishes might develop bottom-up. These bottom-up processes may create momentum for additional HP changes. Since this process of change has been extensively discussed in the process evaluation of HPSF [1], we did not add this elaboration to the text in the current study, but we only included it in the footnote below Figure 1 (see previous comment).

- how did the team get authorization to collect data for BMI score? it considered as medical data, bound by secracy in my country. could the authors elaborate a little bit on this ethical issue?

This study is part of an overall study that investigates HPSF. Ethical approval for the overall study was obtained by the Medical Ethics Committee Zuyderland, located in Heerlen (Parkstad, the Netherlands). All children and their parents had to sign an informed consent form prior to participation. By this form they agree that researchers that are part of the overall study could use the collected data. This means that we were allowed to combine all data in this study and that no new informed consent had to be signed specifically for this study.

- also the mechanism involved in the favorable change in BMI is not presented in the project's logic model. How can it be assumed that this effect results from implementation of the project?

The assumption is that the combination and interaction of all realized HP changes will impact children’s health behaviours, such as their dietary and PA behaviours. A key assumption in this cause-effect relation concerns the non-linearity: it is assumed that small changes in a school can produce large effects at a so-called ‘tipping’ point. Improved health behaviours of children should lead to improvements in their health and well-being, which also includes healthier BMI z-scores of children. We included these assumptions in the explanatory text of Figure 1 (see previous comments).

- l 119-121: how long was the timeframe for the previous study? in other words how long did it take to observe changes in the BMI of children?

We agree with the reviewer that this information is missing in the manuscript. We added this as follows:

Page 4, Line 126-134: The effects of the full and partial HPSF after one- and two year follow-up were investigated in two previous studies [32, 33]. Significant favourable intervention effects after one- and two years’ follow-up were found for the full HPSF on children’s dietary behaviours for, among others, school water consumption, lunch intake of vegetables and dairy products. Children’s sedentary time and light PA significantly improved after two years’ follow-up. Almost no significant favourable results on children’s health behaviours were found in the partial HPSF. In addition, results have shown that children’s BMI z-scores in both the full and the partial HPSF significantly decreased after two years’ follow-up. This favourable effect was already significant after one year’s follow-up in the partial HPSF, but not yet in the full HPSF.

- from l 199, what do you mean by "element"?  is a rule on snacks a HP Policy which draws from a whole school approach? what about school Policy and how HP is included within curriculum development and school projects and so on.. 

In the HPSF research, we used the term health-promoting elements for initiatives in the school that potentially add to school-wide health promotion. We investigated whether a school has for example a HP policy on snacks and whether it is only written down or actually used in the school. All different HP policies and the actual use of it, also during, e.g., school projects, have led to an overall score whether HP policy is present and to what extent it is used. We acknowledge that this way of measuring also has its limitations, but it definitely helped us to gain a better understanding of the different HP elements in the schools. We improved the text in the manuscript as follows:

Page 6, Line 218-227: In the HPSF research, we used the term health-promoting elements for initiatives in the school that potentially add to school-wide health promotion. …. Elements regarding policy (n=7) were determined by questions on rules and policy on snacks, lunch, treats, sugar-sweetened beverages, sport and energy drinks, water, and special policy on school events.

- the paragraph starting l 313 is difficult to read. same comment with the next paragraph

We agree with the reviewer that these two paragraphs are not easy to read. We made some changes in the format of the text by writing in italic when presenting data. We think this improved readability. Below an example of how we did this.

Page 15, Line 333-369: Children’s starting position differed as well: the children in S1 had a more favourable mean BMI z-score (S1: mean (standard deviation (SD)) 0.028 (1.00) vs S2: 0.092 (1.02)) and were more physically active (e.g., light PA in S1: (mean % per day (SD)) 32.8% (5.48) and in S2: 30.6% (5.98)).

- if I am not mistaken the argument l 382 to 383 (complex interactions) is one of the points made by the authors to justify the choice of questions and methods. it seems Strange to find it as a conclusion from the study.

We agree with the reviewer and formulated it now more as a statement. We did this as follows:

Page 16, Line 405-406: These varying effects across schools can be seen as the result of the nonlinearity of the system and the interaction of contextual aspects with the HP changes in the school.

Overall, this article is a high quality description of the key issues in the field of health promotion reserach on implementation. The methods described are scientifically sound and the description is consistent with the expected validty criteria in research. The results make a key contribution to the field. I highly recomment this for publication. 

References:

1.     Bartelink N, van Assema P, Jansen M, Savelberg H, Moore G, Hawkings J, et al. Process evaluation of the Healthy primary School of the Future: The key learning points. BMC Public Health. 2019;19(1):698.

Reviewer 2 Report

The aim of the study "The moderating role of the school context on the effects of the Healthy Primary School of the Future" was to explore the moderating role of the school context on the effects of a Dutch health promoting school initiative on children’s health and health behaviours, among four primary schools.

I found this study investigation well written and organized, the presentation clear, and data analyses appropriate to the study purposes. Thus, I have just a few notes that may help strengthen the paper:

Introduction:

Page 2 Line 79-80: I suggest deleting the sentence "This arrow represents the main focus of the current study." It is redundant. 

Materials and methods   Study design:

"All children (aged 4 to 12)…."       What are the Mean and DS???? Added these information in the section.

HP practices of the Teachers:

The author talks about a questionnaire based on previous work by Gevers et al. and
O’Connor et al.    What is the validity? 

Line 192-194 "The questionnaire was based on  the Measurement Instrument for Determinants of Innovations (MIDI), a Dutch validated questionnaire developed by Fleuren et al…."      What is the validity? 

Discussion:

Line 380: "These findings underline our hypothesis…"     In the introduction you don't formulate hypothesis, but questions. So I suggest to restructure in accordance with questions part or to reformulate according hypothesis. 

Please follow APA standard clearly in all aspects!

Author Response

Dear editor,

Thank you for allowing us to submit a revised version of our manuscript. We also want to thank the reviewers for their positive judgements and helpful comments on the manuscript. Below we explain how we incorporated the comments in our revision. In the manuscript, we have marked the changes in yellow.

Reviewer 2

The aim of the study "The moderating role of the school context on the effects of the Healthy Primary School of the Future" was to explore the moderating role of the school context on the effects of a Dutch health promoting school initiative on children’s health and health behaviours, among four primary schools.

I found this study investigation well written and organized, the presentation clear, and data analyses appropriate to the study purposes. Thus, I have just a few notes that may help strengthen the paper:

Introduction:

Page 2 Line 79-80: I suggest deleting the sentence "This arrow represents the main focus of the current study." It is redundant. 

We agree with the reviewer, and we have deleted this sentence.

Materials and methods  

Study design:

"All children (aged 4 to 12)…."       What are the Mean and DS???? Added these information in the section.

We understand what the reviewer means. However, we mentioned ‘aged 4 to 12’ to inform the reader about the age of children in these Dutch primary schools, in which all children were invited for participation. Providing a mean and SD is difficult, since the non-participating children and parents did not sign an informed consent form to use their data. To prevent any misunderstanding, we deleted this information here and have added it to the introduction section.

 Page 2, Line 80-81: The aim of the study was to explore the moderating role of the school context on the effects of HPSF among four primary schools (aged 4 to 12).

HP practices of the Teachers: The author talks about a questionnaire based on previous work by Gevers et al. and O’Connor et al.    What is the validity? 

Line 192-194 "The questionnaire was based on the Measurement Instrument for Determinants of Innovations (MIDI), a Dutch validated questionnaire developed by Fleuren et al…." What is the validity? 

We agree with the reviewer that we could provide more information about the validity of the two questionnaires. We added this information to the manuscript as follows:

Page 6, Line 195-198: The questionnaire was based on previous work by Gevers et al. [40, 41] and O’Connor et al. [42], in which acceptable to good test-retest reliability of their instruments has been found.

Page 6, Line 206-211: The questionnaire was based on the Measurement Instrument for Determinants of Innovations (MIDI), a Dutch questionnaire developed by Fleuren et al. [43]. They have developed it by a systematic review of empirical studies and a Delphi study among implementation experts. The questionnaire has been used in many different implementation studies, especially in the school setting; no specific research has been conducted to evaluate its validity and reliability [29].

Discussion:

Line 380: "These findings underline our hypothesis…"     In the introduction you don't formulate hypothesis, but questions. So I suggest to restructure in accordance with questions part or to reformulate according hypothesis. 

We agree with the reviewer and have reformulated the sentence in the discussion section by changing the word ‘hypothesis’ to ‘rationale’.

Page 16, Line 403-405: These findings underline our rationale that the school context influences the effects of school health promotion efforts.

Please follow APA standard clearly in all aspects!

We thank the reviewer for this comment. We checked the website of IJERPH again and changed the references in accordance to their guidelines.